# Continuous Spatiotemporal Events Decoupling through Spike-based Bayesian Computation

**Yajing Zheng**[1✉]    **Jiyuan Zhang**[1]    **Zhaofei Yu**[1,2✉]    **Tiejun Huang**[1,2]

[1] State Key Laboratory for Multimedia Information Processing,
School of Computer Science, Peking University
[2] Institute for Artificial Intelligence, Peking University
{yj.zheng,yuzf12,tjhuang}@pku.edu.cn
jyzhang@stu.pku.edu.cn

## Abstract

Numerous studies have demonstrated that the cognitive processes of the human brain can be modeled using the Bayes theorem for probabilistic inference of the external world. Spiking neural networks (SNNs), capable of performing Bayesian computation with greater physiological interpretability, offer a novel approach to distributed information processing in the cortex. However, applying these models to real-world scenarios to harness the advantages of brain-like computation remains a challenge. Recently, bio-inspired sensors with high dynamic range and ultra-high temporal resolution have been widely used in extreme vision scenarios. Event streams, generated by various types of motion, represent spatiotemporal data. Inferring motion targets from these streams without prior knowledge remains a difficult task. The Bayesian inference-based Expectation-Maximization (EM) framework has proven effective for motion segmentation in event streams, allowing for decoupling without prior information about the motion or its source. This work demonstrates that Bayesian computation based on spiking neural networks can decouple event streams of different motions. The Winner-Take-All (WTA) circuits in the constructed network implement an equivalent E-step, while Spike Timing Dependent Plasticity (STDP) achieves an equivalent optimization in the M-step. Through theoretical analysis and experiments, we show that STDP-based learning can maximize the contrast of warped events under mixed motion models. Experimental results show that the constructed spiking network can effectively segment the motion contained in event streams.

## 1   Introduction

Bayesian computation is a fundamental concept in statistics, machine learning, and computational neuroscience [35, 3]. The Bayesian brain hypothesis suggests that the brain functions as a probabilistic generative model, simultaneously inferring hidden causes of sensory inputs and refining its model parameters [22, 8, 21]. Bayesian computation requires a generative model to predict observations, formulated as the joint probability $P(x, y)$ of observations $x$ and hidden states $y$. This joint probability can be decomposed into the prior $P(y)$ and the likelihood $P(x|y)$, which are combined using the Bayes theorem to update the before a posterior probability $P(y|x)$.

The brain's inferential processes use probabilistic models to represent and update perceptions based on new sensory input, framing perception as an active test of predictions against sensory data [11, 34].

---

✉ Corresponding authors.

38th Conference on Neural Information Processing Systems (NeurIPS 2024).

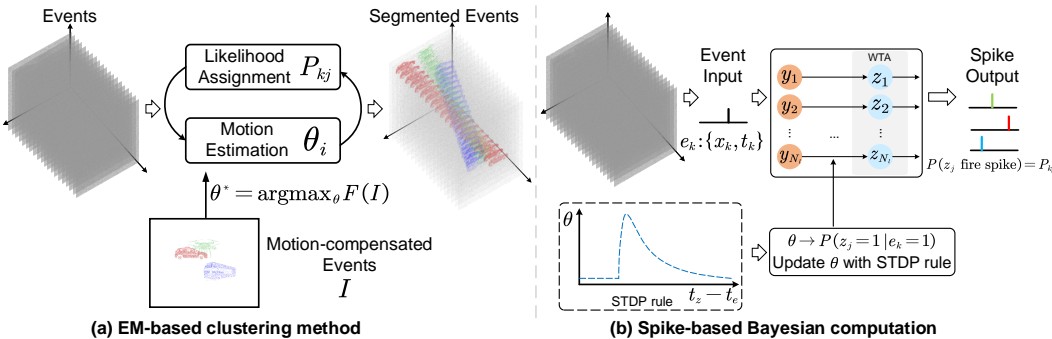

Figure 1: Comparison of EM-based Clustering and the proposed Spike-based Bayesian Computation Methods for Event Decoupling.

Bayesian inference in the brain operates through neural circuits, where spikes encode activity, and synaptic weights facilitate information integration. Soft-WTA circuits, found in cortical microcircuits [7], support selective activation of neurons, allowing the most active neurons to suppress less active ones, which improves computational efficiency. This competition-driven mechanism works in tandem with STDP [4, 9], which strengthens synapses based on precise spike timing, aligning with Bayesian principles to refine predictions. This Bayesian framework has promising applications in brain-machine interfaces and neuromorphic computing [25]. While existing frameworks demonstrate feasibility, their application to complex, spatiotemporal tasks remains a rich area for further exploration.

Recent advances in neuromorphic vision sensors [20, 13, 18] mimic retinal function, generating asynchronous event streams based on light intensity changes. Unlike traditional RGB cameras, these sensors capture motion effectively and reduce blur with high temporal resolution. Event cameras generate events independently at each pixel, complicating data with various object movements and camera motion. This "chicken-and-egg" problem of distinguishing and solving different motions can be addressed using a Bayesian framework combined with the EM algorithm [1], iteratively updating motion model probabilities.

Previous work [38, 44] has demonstrated that jointly optimizing motion parameters and updating event-cluster membership probabilities in an EM fashion can solve motion segmentation problems. This approach achieves per-event segmentation rather than rough region segmentation and supports multiple motion models. The optimization of motion parameters is mainly achieved using the Contrast Maximization (CM) algorithm [14], aligning events to a reference time similar to motion compensation in video processing.

*How can an SNN model implementing Bayesian computation decouple and segment event streams generated by different motion patterns?* Earlier spike-based EM algorithms primarily focused on generating models for mixture distributions, such as Gaussian mixture models, where the probability of each sample belonging to each sub-distribution is equal. In contrast, in motion segmentation, the probability of each event stream belonging to a specific motion parameter distribution is variable. Additionally, optimizing for mixture distributions involves maximizing the likelihood function of each distribution given the observed samples, which differs from the objective of maximizing contrast in event-based motion segmentation. Therefore, applying WTA circuits combined with STDP for motion segmentation in event streams presents several challenges.

This work proposes a spike-based Bayesian computation framework for event segmentation. We theoretically prove that the proposed framework is equivalent to previously implemented event-based motion segmentation algorithms via motion compensation. Experimental results demonstrate that the constructed spiking network can effectively segment motions in event streams. This work provides a theoretical foundation for applying the Bayes theorem using spiking neural networks to event stream decoupling tasks. We aim to provide a theoretical foundation for applying Bayesian inference using SNNs for event stream decoupling tasks, leveraging the energy efficiency of SNNs. This is particularly relevant for neuromorphic hardware (NMHW), e.g., Loihi [5], SpiNNaker [12] and Tianjic [6], known for their low latency and power consumption. Neuromorphic computing emulates the brain's low-power yet complex visual task-processing capabilities. Previous research

validated the hypothesis that SNNs can implement Bayesian inference [29]. However, its application to neuromorphic computing hardware remains unverified. Our work aims to validate the capability of a spike-based Bayesian computation framework applied to neuromorphic sensors.

Contributions of this work can be summarized as follows:

- Develop a spiking Bayesian computation framework for continuous motion segmentation in event streams, demonstrating its ability to perform similarly to previous EM algorithm-based models.
- Show that the WTA circuit can implement the E-step and that STDP rules can achieve the M-step for contrast maximization, validating these components' theoretical and practical efficacy.
- Verify the proposed network can effectively learn online from continuous input, enabling accurate motion segmentation through the firing of output neurons.

## 2  Related Works

**Spike-based Bayesian Computation.**   Spike-based Bayesian computation employs various methods to achieve probabilistic inference and optimization in neural circuits, such as maximum likelihood or posterior function. For example, SNNs implement Bayesian inference using belief propagation on binary MRFs and tree-based reparameterization for exponential family distributions [31, 40, 41, 42], which approximate the posterior probability. Spiking neuron-based neural sampling [2, 40] uses a non-reversible process, like Markov chain Monte Carlo (MCMC) sampling. A particularly suitable method for complex tasks like motion segmentation in event streams involves using WTA circuits combined with STDP [29, 28] learning rules to implement the EM algorithm for decoupling mixture distribution. This approach leverages the competitive dynamics of WTA circuits to estimate joint probability distributions, offering high accuracy and robustness in learning and inference. The WTA-STDP framework is well-suited for handling the asynchronous and event-driven nature of spiking data, making it effective for segmenting complex motion patterns in event streams.

**Event-based motion segmentation.**   Recent methods [26, 16, 39, 33, 27, 38] in event-based motion segmentation leverage clustering, probabilistic models, motion compensation, and deep learning. Cluster-based techniques group events by similar motion patterns and are simple to implement, but may struggle with complex scenes. Probabilistic approaches, such as those using Bayesian frameworks and the EM algorithm [38, 44], iteratively estimate motion parameters and provide robust segmentation, though they can be computationally intensive. Motion compensation aligns events to a reference frame to enhance sharpness (contrast) of warped events [38], offering high accuracy, but may be sensitive to parameter initialization. Deep learning methods employ neural networks to predict motion segments directly from raw event streams, providing high performance and adaptability [19, 37, 27], but requiring large datasets for training. Some works [32, 43] also attempt to use SNN for estimating motion in event streams, thereby achieving motion segmentation or object detection. The main idea of these methods is to first convert the event stream into time-event-like inputs, rather than directly distinguishing the event stream in the spatiotemporal dimension. Consequently, these methods exhibit a certain lag and can easily be disturbed by camera self-motion, which interferes with the analysis of moving objects.

## 3  Preliminaries

### 3.1  Event Cameras.

Dynamic Vision Sensors (DVS) [23] in event cameras detect brightness changes independently at each pixel, generating events instead of capturing images at fixed intervals. An event $e_k = (x_k, t_k, q_k)$ occurs when the intensity change $\Delta L(x_k, t_k)$ at pixel $x_k$ exceeds a threshold $\Theta$. The change in intensity is expressed as:

$$\Delta L(x_k, t_k) = L(x_k, t_k) - L(x_k, t_k - \Delta t_k) = q_k \Theta, \tag{1}$$

where $L(x, t)$ is the logarithmic intensity at pixel $x$, $t_k$ is the event timestamp, $\Delta t_k$ is the interval since the last event at the same pixel, and $q_k \in \{-1, +1\}$ indicates the polarity of the intensity change. Since our work does not use event polarity, this property will not be referenced further. This

asynchronous mechanism enables event-based cameras to capture dynamic scenes with high temporal resolution and low latency, making them ideal for real-time applications.

## 3.2 Event-based Motion Segmentation in EM Fashion.

In previous event-based motion segmentation algorithms, the probability that an event stream $e$ belongs to different motion models $z$ is denoted as $P$. The motion model that generates the event is then used to warp the event positions, resulting in a sharp Image of Warped Events (IWE). The IWE $I_j(x)$ for a given motion parameter $\theta_j$ is typically calculated by warping events onto a specific time plane, where $x$ represents the pixel location, and $j$ refers to the $j$-th motion model:

$$I_j(x) = \sum_{k=1}^{N_e} p_{kj} \delta(x - x'_{kj}), \tag{2}$$

where $\delta(\cdot)$ denotes the Dirac function, and $p_{kj}$ represents the probability that event $e_k$ belongs to motion model $z_j$. The Dirac function can also be replaced by a smoother kernel function, such as the Gaussian function. $x'_{kj}$ is the transformed event position obtained using motion parameters $\theta_j$:

$$x'_{kj} = W(x_k, t_k; \theta_j). \tag{3}$$

The objective of the entire model is to find the motion parameters $\theta^*$ and the event-cluster probability $\mathbf{P}^*$ that maximize the variance of all IWEs corresponding to different motions:

$$(\theta^*, \mathbf{P}^*) = \arg\max_{(\theta, \mathbf{P})} \sum_{j=1}^{N_\ell} \mathrm{Var}(I_j). \tag{4}$$

In EM fashion, we first perform the E-step to calculate the posterior probabilities of the samples belonging to the distribution based on the assumed model and parameters. The initial motion parameters are used to calculate the IWE for different motions. The responsibility $p_{kj}$ for each event belonging to a motion model is then computed using the formula:

$$p_{kj} = \frac{I_j(x'_{kj}(\theta_j))}{\sum_{i=1}^{N_\ell} I_i(x'_{kj}(\theta_i))}. \tag{5}$$

Next, in the M-step, these responsibility values are used to optimize the motion parameters, maximizing the objective function $f(\theta)$. In this work, we use a different variance form to [38], but the effect is the same. The objective function is computed as:

$$f(\theta) = \sum_{j=1}^{N_\ell} \mathrm{Var}(I_j) = \sum_{j=1}^{N_\ell} \mathbb{E}_j[x^2] - \mathbb{E}_j[x]^2, \tag{6}$$

where $\mathbb{E}[x]$ represents the expectation process as:

$$\mathbb{E}_j[x] = \frac{1}{|\Omega|} \int_\Omega I_j(x) \, dx, \tag{7}$$

$$\mathbb{E}_j[x^2] = \frac{1}{|\Omega|} \int_\Omega I_j(x)^2 \, dx. \tag{8}$$

$\Omega$ denotes the image plane. The gradient of the variance $\mathrm{Var}(I_j)$ is given by:

$$\Delta\theta_j = \frac{\partial \mathrm{Var}(I_j)}{\partial \theta} = \frac{\partial \mathbb{E}_j[x^2]}{\partial \theta} - 2\mathbb{E}_j[x]\frac{\partial \mathbb{E}_j[x]}{\partial \theta}, \tag{9}$$

This method effectively distinguishes event streams generated by different motions and produces a sharp IWE. Other optimization strategies, such as the conjugate gradient method [30], can be applied to update the motion parameters $\theta_j$.

## 4 Spike-based Bayesian Computation for Motion Segmentation

The main procedure for event-based motion segmentation involves three key steps: 1) obtaining the IWE, 2) assigning probabilities for events corresponding to different motion models, and 3) optimizing the motion parameters to maximize the contrast of the IWE. In our model, we use a network with a WTA circuit to compute the IWE and the probabilities $P$, and we update the motion parameters using the STDP rule. Below, we will demonstrate that our constructed model and learning method approximates the event-based motion segmentation approach.

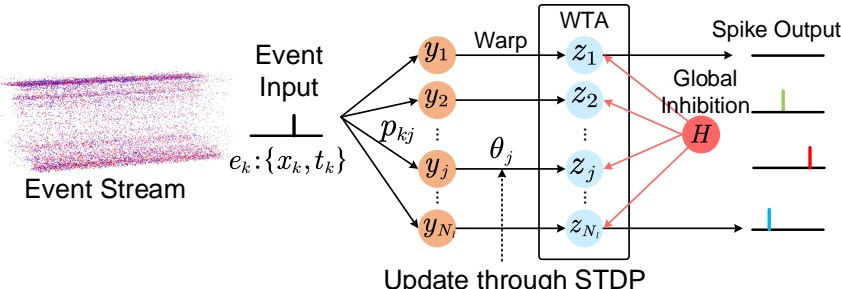

Figure 2: Architecture of the spike-based motion-segmentation network.

## 4.1 Model Construction and Learning

**E-step.** The first step of event-based motion segmentation involves obtaining the IWE for different motion parameters and then determining the event-cluster responsibility values $p_{kj}$ by comparing the contrast of the IWE at the corresponding positions after event warping. This operation is analogous to a neuron competition, where the motion model that produces the highest contrast gains ownership of the events mapped to that location. This process can be implemented using a WTA circuit, which is how our proposed network updates the responsibility values for different events.

As shown in Fig. 2, in our network, neurons $y$ representing different motions $j$ are responsible for warping events and transmitting them to the output neurons $\mathbf{z}$. There is a one-to-one correspondence between $y$ and $\mathbf{z}$. The output neuron $\mathbf{z}_j$ follows an integrate-and-fire (IF) model [15], with the membrane potential $\mathbf{u}_j$ expressed as:

$$\mathbf{u}_j(t) = \sum_{k=1}^{N_e} W_j(e_k, p_{kj}; \theta_j),\tag{10}$$

where $W_j$ represents the operation of the motion neuron $y_j$. It is evident that by accumulating the warped event streams, equivalent to obtaining the IWE in Eq. 2.

The output neuron $\mathbf{z}_j$ receives feedback from a global inhibition neuron $\mathcal{H}$. The value of the global inhibition neuron is the sum of all IWE values, i.e., $\mathcal{H}(t) = \sum_j \mathbf{u}_j(t)$. Here, we use a stochastic firing model for $\mathbf{z}_j$, where the firing probability depends on the membrane potential in conjunction with the inhibition neuron $H$. This computation can be formulated as:

$$p(\mathbf{z}_j \text{ fire at time } t) = \frac{\mathbf{u}_j(t)}{\mathcal{H}(t)}.\tag{11}$$

Unlike the WTA circuits constructed by Nessel et al. [29, 28], our output neurons form a tensor corresponding to the event space dimensions. The synaptic parameters updated by STDP do not represent the connection weights between scalar neurons but rather the coefficients of functions representing the receptive fields of different motion neurons (e.g., functions that linearly transform along the direction of motion). The concept of neurons representing tensor values has also been applied in various works, such as in capsule networks [36].

**M-step.** After obtaining the network's output, we can update the network parameters using the STDP rule based on the relationship between the firing of $\mathbf{z}_k$ and the input events. In our previous definitions, the network parameters include only the motion parameters $\theta_j$. The probability $p_{kj}$ is derived from the firing probability of the output neurons and is also used as the input for the next step in the network training process.

According to the STDP rule, if the firing of the presynaptic neuron leads to the firing of the postsynaptic neuron, their synaptic weight is increased, corresponding to long-term potentiation (LTP). Otherwise, the synaptic weight is decreased, corresponding to long-term depression (LTD). The synaptic weight is updated as follows:

$$\theta \to p(\text{presynaptic neuron fired within } [t' - \sigma, t'] \mid \text{postsynaptic neuron fires at } t').\tag{12}$$

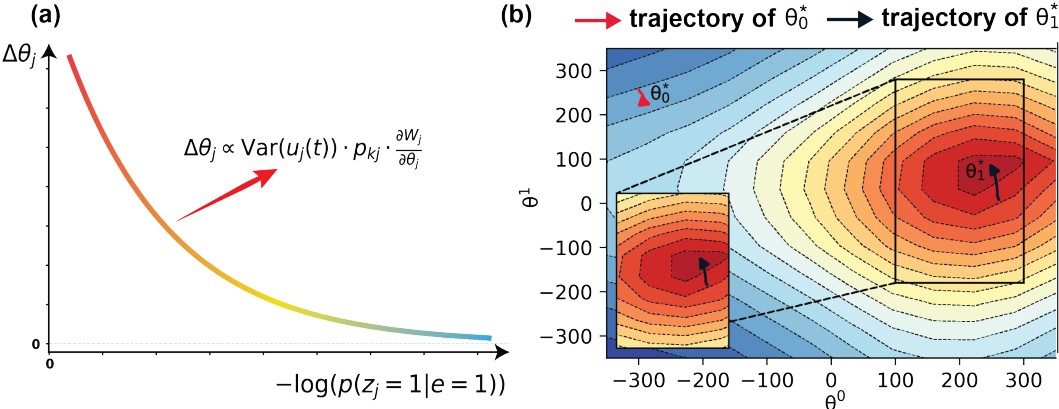

Figure 3: Illustration of learning through STDP. **(a).** Learning curves of STDP for motion parameters $\theta$. **(b).** Optimization trajectory of parameters during STDP learning. The heat map shows the gradient of $f(\theta)$ in Eq. 6 as motion parameters change.

In this context, the probability of firing between the presynaptic event $e$ and the postsynaptic neuron $\mathbf{z}$ is positively correlated with their association probability $p_{kj}$. Consequently, the update direction of the motion parameter $\theta_j$ is also positively correlated with $p_{kj}$. Combining this with the concept of motion compensation, the goal of updating $\theta_j$ is to maximize contrast $\mathrm{Var}(I_j)$, so the gradient direction is related to the contrast of $\mathbf{u}_j(t)$ (where $\mathbf{u}_j(t)$ is equivalent to IWE $I_j$). Therefore, in our proposed network, the gradient update for the motion parameter $\theta_j$ is given by:

$$\Delta\theta_j = \eta \cdot \mathrm{Var}(\mathbf{u}_j(t)) \cdot p_{kj} \cdot \frac{\partial W_j}{\partial \theta_j}. \tag{13}$$

Fig. 3(a) shows the learning curve of STDP. We do not explicitly show the use of LTD for updates because the presence of $p_{kj}$ in the update formula ensures that if the event stream does not belong to the motion model, it will not significantly affect the update of its motion parameters. This approach is reasonable, as a low $p_{kj}$ value indicates that the current input event stream does not maximize the contrast for the motion model. Hence, it should not influence the learning of the motion neuron parameters, thereby preventing interference with the detection of other motion patterns in the event stream.

Our objective is to maximize the sum of the variances of the IWE across different motion parameter distributions (Eq. (4)). As described in Fig. S7 of the Appendix, the variance indicates that correct motion patterns concentrate the event flow distribution along the object's edges, resulting in higher variance. In contrast, incorrect motion patterns disperse the event flow, leading to lower variance. In the network we designed, which includes a WTA mechanism, the parameters of the motion neurons, denoted as $\theta$, are optimized using the STDP rule to achieve this goal. As shown in Eq. (12) and Fig. 3, when $P(z_j = 1|e_k = 1)$, only event streams that match motion pattern $j$ will activate the corresponding motion neuron to adjust its parameters, thereby maximizing the variance of IWE (encoded by $u$) associated with motion pattern $j$.

In our network, applying STDP to optimize the parameters of motion neurons does not strictly follow the gradient of the objective function for each parameter. However, the angle between the update direction and the gradient of the objective function is less than 90 degrees. Previous work has demonstrated that synaptic updates need only maintain an angle less than 90 degrees with the error function to achieve optimization [24]. We will also show that under this update rule, the contrast of the IWE can gradually increase.

### 4.2 Equivalence of STDP-Rule Updates to M-Step in Event-Based Motion Segmentation

In the M-step of event-based motion segmentation, the goal is to optimize the parameter values by maximizing the contrast of the IWE. The objective is to increase the contrast of all IWE values during the optimization process, and various optimization strategies can be employed. Our proposed STDP rule effectively increases the value of $\mathrm{Var}(\mathbf{u}_j)$.

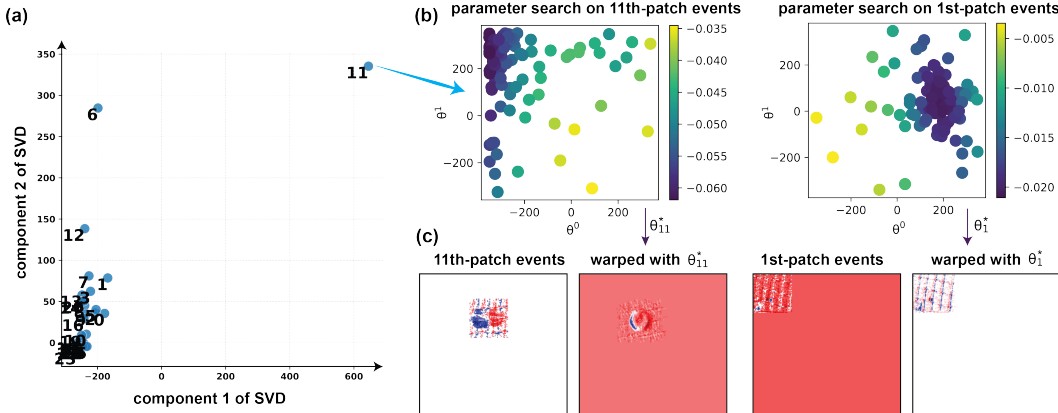

Figure 4: Initialization of parameters $\theta$ through sampling parameters with the contrast of IWE as the objective function. **(a).** SVD components of parameters of different patches. **(b).** Sampling process of parameters of different patches. **(c).** Warping events with the best sampling parameters $\theta^*$.

The variance formula is expanded as follows:

$$\mathrm{Var}(\mathbf{u}_j) = \frac{1}{N} \sum_{k=1}^{N} \mathbf{u}_j(k)^2 - \left( \frac{1}{N} \sum_{k=1}^{N} \mathbf{u}_j(k) \right)^2 , \tag{14}$$

where $N = |\Omega|$. Assume that after the STDP update, the parameter becomes $\theta'_j = \theta_j + \Delta\theta_j$, resulting in a new output $\mathbf{u}'_j$ with variance $\mathrm{Var}(\mathbf{u}'_j)$.

The updated variance formula:

$$\mathrm{Var}(\mathbf{u}'_j) = \frac{1}{N} \sum_{k=1}^{N} \mathbf{u}'_j(k)^2 - \left( \frac{1}{N} \sum_{k=1}^{N} \mathbf{u}'_j(k) \right)^2 . \tag{15}$$

The updated output is :

$$\mathbf{u}'_j(k) = \mathbf{u}_j(k) + \Delta\mathbf{u}_j(k), \tag{16}$$

where $\Delta\mathbf{u}_j(k)$ is the change induced by $\Delta\theta_j$. The change in variance is calculated as:

$$\mathrm{Var}(\mathbf{u}'_j) - \mathrm{Var}(\mathbf{u}_j) = \frac{1}{N} \sum_{k=1}^{N} (\mathbf{u}_j(k) + \Delta\mathbf{u}_j(k))^2 - \left( \frac{1}{N} \sum_{k=1}^{N} (\mathbf{u}_j(k) + \Delta\mathbf{u}_j(k)) \right)^2 - \mathrm{Var}(\mathbf{u}_j). \tag{17}$$

Since $\Delta\mathbf{u}_j(k)$ is a small change relative to $\theta_j$, we can use a first-order approximation:

$$\Delta\mathbf{u}_j(k) \approx \frac{\partial \mathbf{u}_j(k)}{\partial \theta_j} \cdot \Delta\theta_j. \tag{18}$$

The parameter update increases the variance. According to the STDP update rule, the direction of $\Delta\theta_j$ is consistent with $\partial\mathrm{Var}(\mathbf{u}_j)/\partial\theta_j$. Therefore, the variance will increase after the update.

Fig 3(b) illustrates the optimization trajectory of the motion parameters when applying STDP to maximize the objective function $\mathrm{Var}(\mathbf{u}_j(t))$ for motion segmentation. The results show that under the STDP learning rule, influenced by the WTA circuit, the input events $e$ most relevant to the motion are used for optimization. This allows $\theta_1$ to progress in a direction that maximizes the contrast of the corresponding event stream. Additionally, this process does not interfere with the learning of other motion patterns $\theta_0$. The WTA circuit combined with the STDP rule ensures that the contrast of event streams not belonging to the motion pattern is not forcefully maximized.

# 5 Experiments

## 5.1 Implementation Details.

**Parameter Initialization.**   The proposed spike-based Bayesian Computation framework and its corresponding event-based clustering framework are both locally convergent algorithms. The STDP rule adjusts the weights based on the firing patterns of pre- and post-synaptic neurons, and it does not guarantee convergence to a global solution. To ensure both algorithms converge to a better solution, we adopt a strategy similar to the event-based layered algorithm for initializing $\theta$ and $\mathbf{P}$. The number of motion models $N_\ell$ and the type of motion models (e.g., linear motion, affine, or rotation) are hyperparameters that need to be predefined. Considering the ultra-high temporal resolution advantage of event cameras, we primarily set the motion model to linear motion, i.e., $\theta = \{v_x, v_y\}$. After setting the number of motion models $N_\ell$ and the form of $\theta$, we take a subset of events for parameter initialization.

Specifically, we divide the events into different patches and use the specified motion parameters to maximize the contrast of the IWE corresponding to these patch events. To search for parameters more efficiently instead of using brute force methods (e.g., grid search), we employ a combination of random sampling and Bayesian optimization using the Tree-structured Parzen Estimator (TPE) [10]. After completing the parameter search, we obtain a parameter set corresponding to the number of patches $N_{np}$ ($N_{np} > N_\ell$). Since different patches may have similar motion parameters, we use Singular Value Decomposition (SVD) [17] to analyze the components of the returned parameter set and select the top $N_\ell$ parameters with the most significant differences as the initialization parameters for the algorithm.

Fig. 4 shows the parameter initialization process in a sequence from the Extreme Event Dataset (EED) [26], which includes both camera self-motion and a moving object. The initial parameter search identifies two significantly different motion parameters corresponding to the event streams from the 1st- and 11th- patches, respectively.

**Network Learning and Inference.**   After initializing the parameters, we select a fixed number of events in chronological order, dividing all events into different packets $\{e^n\}_{n=1}^{N_g}$ as inputs to the network over time. During online learning, we also split the $n$-th events packet $\{e^n\}$ into different patches and feed them into the network. After optimizing the parameters $\theta$ for several epochs, we obtain the optimized motion parameters $\theta^*$, and then input all events into the network to get the responsibilities $\mathbf{P}$ of all events belonging to different motion parameters. The motions estimated by clustering $\{e^n\}$ can be propagated in time to predict an initialization for the clusters of the next packet $\{e^{n+1}\}$. All steps of the proposed method are summarized in Algorithm. 1.

---

**Algorithm 1** Spike-based Bayesian Computation for Event Motion Segmentation

---

**Input:** Events packet $\{e_k^n\}_{k=1}^{N_e}$, number of clusters $N_\ell$.
**Output:** Cluster parameters $\theta = \{\theta_j\}_{j=1}^{N_\ell}$, event-cluster assignments $\mathbf{P}$.

1: **procedure**
2:     Initialize the unknowns $(\theta, \mathbf{P})$ by sampling potential motion patterns of different patches of events based on the TPE [10] and SVD [17].
3:     **while** not converged **do**
4:         **E-step** Compute the event-cluster assignments $p_{kj}$ based on the WTA circuit using Eq. 10 and Eq. 11.
5:         **M-step** Update the motion parameters of all clusters using Eq. 13 based on the STDP rule.
6:     **end while**
7: **end procedure**

**Network Learning and Inference:**

1: Divide events into packets $\{e^n\}_{n=1}^{N_g}$ based on fixed event count.
2: **for** each packet $e^n$ **do**
3:     Split events into patches and feed into the network.
4:     Optimize parameters $\theta$ for several epochs to obtain optimized motion parameters $\theta^*$.
5:     Feed all events into the network to obtain event responsibilities $\mathbf{P}$.
6: **end for**

---

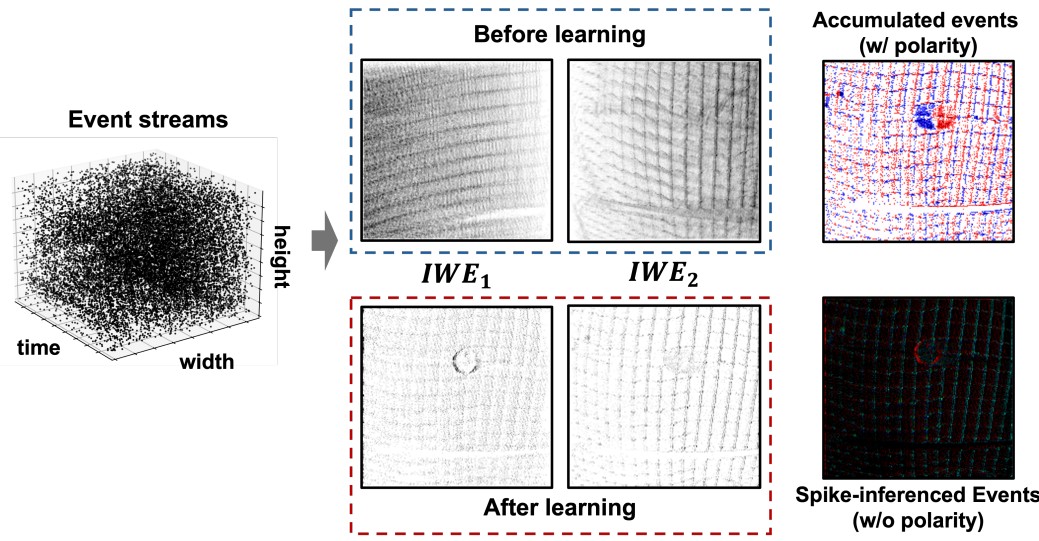

Figure 5: Examples of motion segmentation through the proposed spike-based EM models.

Our model is event-driven and operates with parallel computations across different patches. The primary focus is on CPU-based verification due to minimal graphical operations, ensuring fast processing as an online learning algorithm. Additionally, the speed on both GPU and CPU is comparable. Our network structure is compact, resulting in low memory consumption.

## 5.2 Evaluation on Event Motion Segmentation Datasets

Fig. 5 shows the optimized motion parameters $\theta^*$ and the spike firing rates $p_{kj}$ of the output neurons $\mathbf{z}$ for the scene described in Fig. 4. The figure also presents the IWEs and the events warped and fused according to the corresponding motion parameters. After training with the proposed network, the IWEs for different motion parameters show higher contrast compared to before learning. This allows for effective separation of the camera's self-motion from the motion of the ball. The Spike-inference Events in Fig. 5 are accumulated based on the spike activity of different neurons represented by colors. To verify that the proposed spiking neural network can learn the parameters of motion neurons online from continuous input, we continuously feed the event stream into the network and observe the firing state of the output neurons $\mathbf{z}$.

Fig. 6 shows the network's performance in segmenting three scenarios in the EED that involve mixed camera self-motion and high-speed moving objects. In these scenes, the background event streams vary in density and shape. Despite these varying backgrounds, the output neurons can still distinguish the motion parameters of moving objects. This demonstrates that the proposed network can learn to suppress irregular input spike patterns and, through local plasticity learning, identify the motion parameters that maximize contrast in specific regions, thereby accurately locating different moving objects.

## 6 Conclusions & Discussions

This paper proposes a spike Bayesian computation framework for continuous motion segmentation in event streams. We demonstrate that the constructed network can achieve the same effect as previous event-based motion segmentation models using the Expectation-Maximization (EM) algorithm. Specifically, the WTA circuit in the network implements the equivalent E-step, and the STDP rule for adjusting network parameters realizes the equivalent M-step for contrast maximization. Both theoretical analysis and experimental results show that STDP-based learning can optimize the contrast of mapped images under a mixture motion parameter model. Using the Extreme Event Dataset, we validate the network's ability to learn online from continuous input and perform motion segmentation through the firing of output neurons.

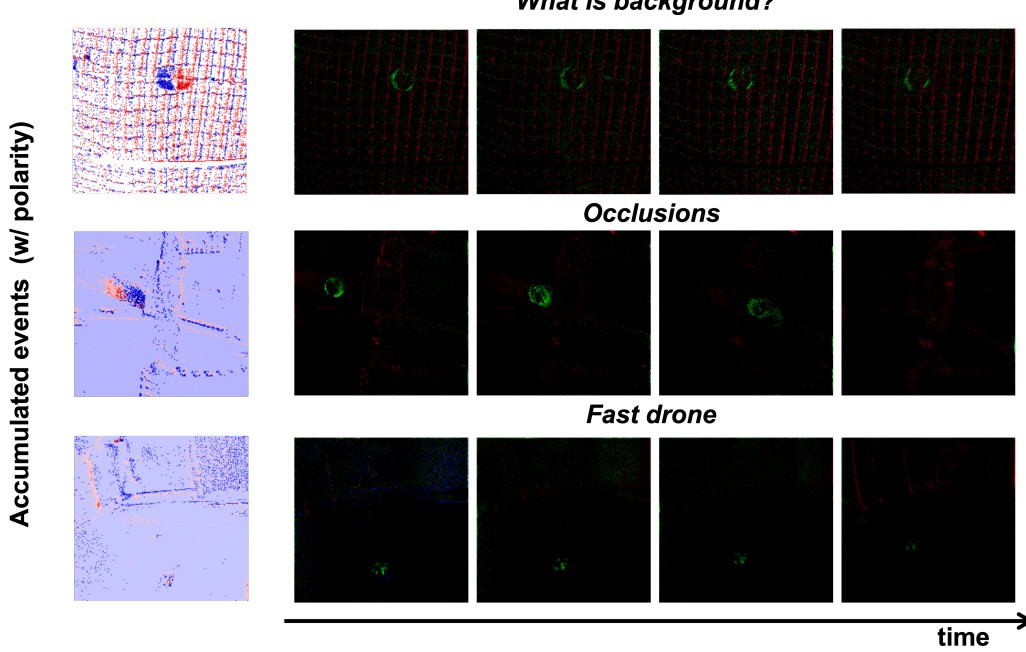

Figure 6: Motion segmentation results for continuous event streams. Different colors represent the firing of different output neurons **z** of the proposed spike-based network.

**Limitations.** This work primarily aims to demonstrate that a biologically inspired network framework implicitly does Bayesian computation can decouple spatiotemporal data, proposing a prototype framework applicable to neuromorphic cameras. The validation focuses on motion segmentation in event streams rather than pursuing state-of-the-art performance. Therefore, there is limited quantitative performance evaluation and comparison with existing event-based motion segmentation algorithms. This limitation arises because the network relies on local learning rules and may not converge to the optimal solution, being highly dependent on parameter initialization.

## Acknowledgments

This work was supported by the National Natural Science Foundation of China (62306015, 62176003, 62088102), the China Postdoctoral Science Foundation (2023T160015), the Young Elite Scientists Sponsorship Program by CAST (2023QNRC001), and the Beijing Nova Program (20230484362).

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

# A  Appendix

## A.1  Explanation of $Var(I_j)$

As shown in Fig. S7, correct motion models concentrate event distributions at object edges, resulting in higher variance, while incorrect models disperse events, leading to lower variance. This helps in validating the accuracy of motion pattern detection.

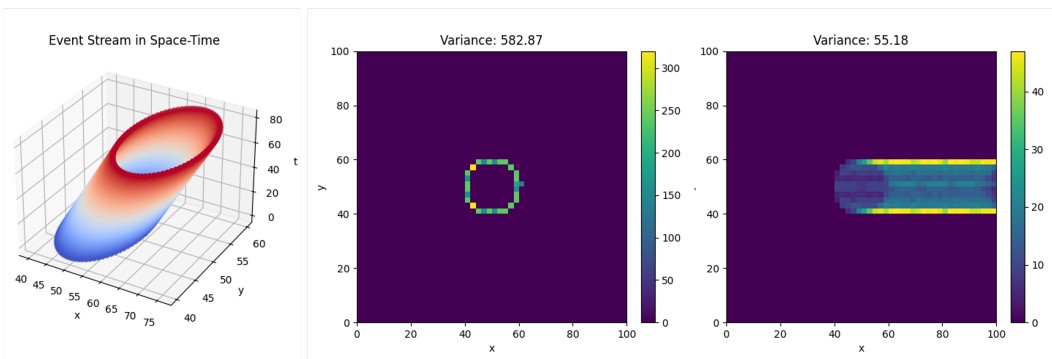

Figure S7: Explanation of $Var(I_j)$. From left to right: events, IWEs of different motion patterns.

## A.2  Joint probability density

From the perspective of a probabilistic model, we can consider that events conforming to different motion patterns emerge over time. However, in the task of dividing event streams based on motion patterns, it is hard to directly generate a model (e.g., a mixture distribution model) using a generative model. In previous work [34], the authors adopted a method based on Eq. (2), which is more suitable for traditional mixture distribution models (e.g., fuzzy mixture models and k-means clustering) to divide event streams by motion.

In fuzzy mixture density and k-means methods, the motion-compensated IWEs do not include the event cluster associations $P$, which means that sharper object boundaries appear in some IWEs compared to others. The key difference between the EM model in this paper and traditional mixture distribution models lies in the fact that not all motion parameters are mixed. Instead, a specific one-to-one relationship is established between the event $e_k$ and the motion neuron $z_j$, resulting in a more precise correspondence.

Therefore, we can define the joint probability distribution as:

$$P(e_k, z_k \mid \theta) = P(z_k \mid \theta) \cdot P(e_k \mid z_k, \theta)$$

where the definition of $P(e_k \mid z_k, \theta)$ relates to the IWE value.

Specifically, we can define the conditional probability of event $e_k$ belonging to motion pattern $z_j$ as:

$$P(e_k \mid z_j, \theta) \propto I_j(x'_{k_j} \mid \theta_j)$$

where $I_j(x'_{k_j} \mid \theta_j)$ represents the IWE value calculated based on the position and time of event $e_k$ using the parameters $\theta_j$ corresponding to motion pattern $z_j$.

## A.3  Motion parameter initialization

In our work, we were inspired by the layered method for event stream motion segmentation described in the EM-based approach and the SOFADS algorithm [s1]. The SOFADS method iteratively refines optical flow estimates through the Track Plane and Flow Plane modules. The Track Plane contains projections of different flow hypotheses, updated based on incoming events. Similarly, we adopted

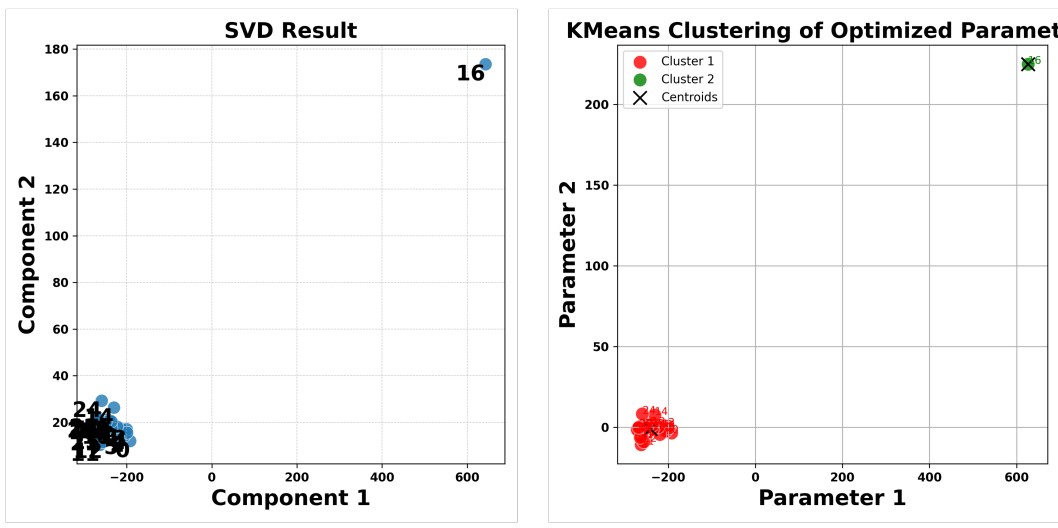

Figure S8: Comparison of selection methods for parameter initialization (SVD vs. KMeans).

a patch-based approach to perform importance sampling on the event stream to identify potential motion parameters based on the optimization objective (contrast of IWE).

Our method involves a search process, with Fig. 4 illustrating this search method combined with SVD analysis. We select the $N_l$ representative parameters with the highest variance as initial values. It is essential to note that the parameter search process is crucial, and the use of SVD can be substituted with K-means for initial parameter selection. As shown in Fig. S8, the parameter points obtained using K-means clustering are similar to those obtained with SVD.

