# OpenReview forum: "Continuous Spatiotemporal Events Decoupling through Spike-based Bayesian Computation"
_NeurIPS.cc/2024/Conference — NeurIPS 2024 poster_

### Official Review · Reviewer_LG4A · 2024-07-09

**Soundness:** 3
**Presentation:** 3
**Contribution:** 3
**Rating:** 7
**Confidence:** 5

**Summary:**

This paper presents a spike-based Bayesian inference framework for motion segmentation with event cameras. By designing neurons that utilize STDP for online learning of motion patterns, the framework can perform the M-step of the EM algorithm in motion segmentation of event streams. Additionally, the WTA circuit implements the E-step, allowing for the online partitioning of event streams into different motion patterns. The authors provide theoretical proof and experimental results to demonstrate the network's spatiotemporal decoupling capabilities for mixed motion patterns of event streams.

**Strengths:**

The authors demonstrate that the SNN framework based on WTA is equivalent to the EM algorithm for motion segmentation of event streams. This online learning approach is compatible with neuromorphic data and beneficial for deployment on low-power, low-latency neuromorphic computing platforms.

• The work is based on the Bayesian brain hypothesis, using a more physiologically interpretable SNN for Bayesian inference. Applying this to spatiotemporal data from neuromorphic cameras represents a promising research direction.

**Weaknesses:**

• The experimental results lack quantitative evaluations. Can the authors further perform object detection and tracking based on the motion segmentation, providing metrics such as object detection success rates and comparisons with other methods?

• The proposed algorithm lacks the analysis of time complexity or processing speed. Can it leverage the low-latency advantage of event cameras?

**Questions:**

Please see weaknesses.

**Limitations:**

There is a need for quantitative evaluations and an assessment of the dependency on parameter initialization.

---

> ### Author Rebuttal · Authors · 2024-08-07
>
> We appreciate your constructive suggestions and have supplemented our work with further results on object detection based on motion segmentation. Specifically, we calculated the detection success rate on the EED dataset, corresponding to Fig. 6 in the main text. Our detection success rates across three test scenarios are `100%`, comparable to many current state-of-the-art algorithms ([s1], [s2]).
>
> **References:**
>
> [s1] Kepple, Daniel R., et al. "Jointly learning visual motion and confidence from local patches in event cameras." Computer Vision–ECCV 2020: 16th European Conference, Glasgow, UK, August 23–28, 2020, Proceedings, Part VI 16. Springer International Publishing, 2020.
>
> [s2] Zheng, Yajing, et al. "Spike-based motion estimation for object tracking through bio-inspired unsupervised learning." IEEE Transactions on Image Processing 32 (2022): 335-349.
>
> **2. Parameter Efficiency and Computational Resource Requirements:**
>
> Our method requires minimal parameters and computational resources. Specifically, the model uses neurons corresponding to different motion models and a single global inhibitory neuron to perform WTA. This parameter efficiency and low computational requirement make our approach particularly advantageous for hardware or neuromorphic hardware deployment. It can fully leverage the low latency and low power consumption characteristics of neuromorphic cameras and computing platforms.
>
> Thank you once again for your invaluable feedback. We hope these additions and clarifications address your concerns and further demonstrate the robustness and potential of our proposed method for event-based motion segmentation and object detection using spiking neural networks.

---

> > ### Comment · Reviewer_LG4A · 2024-08-13
> > **Comments**
> >
> > The authors responded to my earlier questions about the quantitative comparison of their proposed method with SOTA and addressed my concerns regarding delay and time complexity. They provided detection success rates (100%) for the three extreme scenarios in Fig. 6, which are comparable to SOTA methods, but with significantly lower algorithm complexity. After reviewing the authors' replies to all the reviewers, I'm convinced that incorporating the reviewers' feedback, such as suggestions on WNgv descriptions, will greatly enhance the potential of the spike-based Bayesian Computation framework presented in this paper. This framework has enormous potential for low-latency, low-power applications in linking neuromorphic sensors and NMHW. Therefore, I am considering giving it a higher rating.

---

> > > ### Author Response · Authors · 2024-08-13
> > >
> > > Thank you for your valuable feedback! We're glad our responses addressed your concerns, and we appreciate your consideration of a higher rating. We will continue to refine our work based on the reviewers' suggestions.

---

### Official Review · Reviewer_wrk7 · 2024-07-10

**Soundness:** 3
**Presentation:** 3
**Contribution:** 3
**Rating:** 7
**Confidence:** 5

**Summary:**

This work proposes a spike Bayesian computational framework for continuous motion segmentation in event streams and demonstrates that the constructed network can implement an EM-based event stream motion segmentation model. The proposed model uses WTA circuits in the network to achieve an equivalent E-step, while the STDP rules for an M-step for contrast maximization. Experimental results demonstrate the network's online learning effectiveness for continuous inputs on extreme event camera datasets.

**Strengths:**

The proposed network's effectiveness for motion segmentation has been validated on event datasets featuring challenging scenarios that involve mixed camera self-motion and high-speed moving objects. The proposed spike Bayesian inference framework is highly interpretable and applicable to various neuromorphic vision chips and computing hardware, representing a promising research direction.

**Weaknesses:**

The authors mainly use SVD to find different patches' motion patterns for initialization. Why is this method used, and can other methods be employed for selection? It is recommended that the authors conduct ablation experiments to explore further.

**Questions:**

This method primarily targets optical flow motion estimation. For more complex motion patterns, how to design the parameters? How robust is this method against noise in the evaluation of such motion models? The authors should clarify it.

**Limitations:**

The authors have stated the limitations.

---

> ### Author Rebuttal · Authors · 2024-08-07
>
> Thank you for your acknowledgment of our validation of event datasets featuring challenging scenarios, including mixed camera self-motion and high-speed moving objects, which is highly valued. We are pleased that you find our spike Bayesian inference framework to be **highly interpretable** and **applicable to various neuromorphic vision chips and computing hardware**. Your appreciation of our work's potential as a **promising research direction** is encouraging.
>
> ### **Detailed Response**
>
> **1. Motion Parameter Initialization:**
>
> In our work, we were inspired by the layered method for event stream motion segmentation described in the EM-based approach and the SOFADS algorithm [s1]. The SOFADS method iteratively refines optical flow estimates through the Track Plane and Flow Plane modules. The Track Plane contains projections of different flow hypotheses, updated based on incoming events. Similarly, we adopted a patch-based approach to perform importance sampling on the event stream to identify potential motion parameters based on the optimization objective (contrast of IWE).
>
> Our method involves a search process detailed in lines 226-234 of our paper, with Fig. 4 illustrating this search method combined with SVD analysis. We select the $N_l$ representative parameters with the highest variance as initial values. It is essential to note that the parameter search process is crucial, and the use of SVD can be substituted with K-means for initial parameter selection. As shown in Fig. S2 `(please see in the supplementary PDF)`, the parameter points obtained using K-means clustering are similar to those obtained with SVD.
>
> **2. Types of Motion Neuron Parameters:**
>
> The parameters of motion neurons in our framework can accommodate much more complex motion patterns by simply modifying the neuron model. For instance, our model can be extended to handle rotational motion [s2], 4-DOF motion [23], or even 8-DOF homographic motion [s3]. This flexibility underscores the versatility of our framework in adapting to various motion types.
>
> **3. Noise Handling:**
>
> Regarding noise, its randomness means it generally does not conform to any initialized motion model. During motion segmentation, noise tends to have similar probabilities across different categories, effectively filtering it out. This inherent filtering capability enhances the robustness of our segmentation method in real-world scenarios where noise is prevalent.
>
> Thank you once again for your valuable feedback. We hope these clarifications address your concerns and further illustrate the robustness and potential of our proposed method for motion segmentation using event-based neural networks.
>
> **References:**
>
> [s1] Stoffregen, Timo, and Lindsay Kleeman. "Simultaneous optical flow and segmentation (SOFAS) using dynamic vision sensor." arXiv preprint arXiv:1805.12326 (2018).
>
> [s2] Guillermo Gallego and Davide Scaramuzza, “Accurate angular velocity estimation with an event camera,” IEEE Robot. Autom. Lett., vol. 2, no. 2, pp. 632–639, 2017.
>
> [s3] Guillermo Gallego, Henri Rebecq, and Davide Scaramuzza, “A unifying contrast maximization framework for event cameras, with applications to motion, depth, and optical flow estimation,” in IEEE Conf. Comput. Vis. Pattern Recog. (CVPR), pp. 3867–3876, 2018.

---

> > ### Comment · Reviewer_wrk7 · 2024-08-09
> >
> > Thank the authors for the clarifications on the motion pattern initialization and noise handling. I have no more questions.

---

> > > ### Author Response · Authors · 2024-08-12
> > >
> > > Thank reviewer for the valuable feedback and taking the time to review our work. We're glad the clarifications were helpful!

---

### Official Review · Reviewer_pCA1 · 2024-07-11

**Soundness:** 2
**Presentation:** 2
**Contribution:** 2
**Rating:** 5
**Confidence:** 2

**Summary:**

The paper proposes to address motion segmentation at very high temporal resolution via an event-based or spiking implementation of expectation-maximization in a generative model. It demonstrates the performance of the resulting spiking neural networks on example experiments.

**Strengths:**

The strength of the paper is its deep engagement with the spiking neural network literature, as well as its use of spiking networks for the specific type of problem to which they are most suited: event-based computation.

**Weaknesses:**

The paper's major weakness is its lack of clarity, which the authors have discussed and addressed in the review period.

**Questions:**

The authors have addressed my questions, though I would still like to see discussion of how this framework for EM in spiking networks could be generalized beyond motion detection.

**Limitations:**

I am not confident that I can identify the specific limitations of this paper as opposed to the limitations of spiking neural networks generally.

---

> ### Author Rebuttal · Authors · 2024-08-07
>
> We appreciate the reviewer's recognition of our approach to high-resolution motion segmentation using an event-based implementation of the EM algorithm and acknowledge **our deep engagement with the spiking neural network literature**.
> Here, we aim to provide further clarification on the EM framework and our model's specific application.
>
> ### **Clarification on the EM Algorithm Framework**:
>
> The EM algorithm is a versatile framework used to find parameters in probabilistic models with latent variables. While traditionally, the M-step in the EM algorithm maximizes the likelihood function, it can also maximize other objective functions, such as the Evidence Lower Bound (ELBO) [s1] in variational inference or the Penalized Likelihood [s2] in sparse regression models like Lasso regression.
>
> In previous work on event stream motion segmentation [34], the M-step focused on optimizing the contrast of different IWEs, effectively projecting the event distribution variance using motion parameters. Our current approach uses the EM framework to iteratively perform the E-step and the M-step to refine the estimates of motion parameters, achieving event stream motion segmentation. `The relationship between them is shown in Fig. 1 of the main text`. Therefore, our method does not involve the common concept of maximizing log densities in mixture distribution models.
>
> **References:**
>
> [1s] Blei, D. M., Kucukelbir, A., & McAuliffe, J. D. (2017). Variational Inference: A Review for Statisticians. Journal of the American Statistical Association, 112(518), 859-877.
>
> [2] Friedman, J., Hastie, T., & Tibshirani, R. (2010). Regularization Paths for Generalized Linear Models via Coordinate Descent. Journal of Statistical Software, 33(1), 1-22.
>
> ### **Model's Nature and Application:**
>
> Thank you for pointing out the potential confusion regarding our model's nature. *Our proposed approach is not a generative model. Instead, it is designed to learn motion parameters embedded in event streams generated by different motions.* These learned motion parameters help segment the event streams according to their respective motion distributions. The term "motion distribution" here refers to a projection plane where the variance is maximized if the events follow the motion distribution.
>
> By implementing this EM-based motion segmentation using SNNs, we can leverage the low-latency and low-power characteristics of **neuromorphic hardware (NMHW)**, such as `Loihi and Spinnaker`. This combination enhances the performance of event-based cameras and neuromorphic computing systems. We recognize the importance of this aspect and will include more detailed discussions in future revisions of our paper.
>
> Thank you again for your comments. We hope this response clarifies our approach and addresses your concerns comprehensively.

---

> > ### Comment · Reviewer_pCA1 · 2024-08-11
> > **I must be misunderstanding something.**
> >
> > I thank the authors for their close technical engagement here, and would request that they guide me through just a couple more steps of reasoning.
> >
> > I understand that the EM algorithm is ultimately a coordinatewise optimization of the ELBO, first on latent variables and then on model parameters (alternating until convergence).  But what objective, then, are the authors claiming their application of EM optimizes?
> >
> > Similarly, I understand that the authors are claiming to perform EM rather than implement a generative model, but EM is an inference algorithm for probabilistic generative models with latent variables.  If you cannot write down a joint probability density, that is a generative model, to parameterize the ELBO, then the EM algorithm, as I understand it, cannot be used for inference.
> >
> > Could the authors help me through my confusion here?

---

> > > ### Author Response · Authors · 2024-08-12
> > >
> > > We sincerely appreciate the reviewer’s comments and the opportunity to further clarify our work. We are grateful for the chance to provide additional details regarding our optimization goals and the joint probability density.
> > > ### **1. What objective are the authors claiming their application of EM optimizes?**
> > > Our objective is to maximize the sum of the variances of the IWE across different motion parameter distributions (Eq. (4) in the main text). As described in the supplementary PDF (Fig. S1), the variance indicates that correct motion patterns (illustrated by the circles in the middle subfigure) concentrate the event flow distribution along the object's edges, resulting in higher variance. In contrast, incorrect motion patterns disperse the event flow, leading to lower variance. In the network we designed, which includes a WTA mechanism, the parameters of the motion neurons, denoted as $\theta$, are optimized using the STDP rule to achieve this goal. As shown in Eq. (12) and Fig. 3 of the main text, when $P(z_j=1|e_k=1)$, only event streams that match motion pattern $j$ will activate the corresponding motion neuron to adjust its parameters, thereby maximizing the variance of IWE (encoded by $u$) associated with motion pattern $j$.
> > > ### **2. Joint Probability Density**
> > > From the perspective of a probabilistic model, we can consider that events conforming to different motion patterns emerge over time. However, in the task of dividing event streams based on motion patterns, it is hard to directly generate a model (e.g., a mixture distribution model) using a generative model. In previous work [34], the authors adopted a method based on Eq. (2), which is more suitable for traditional mixture distribution models (e.g., fuzzy mixture models and k-means clustering) to divide event streams by motion.
> > >
> > > In fuzzy mixture density and k-means methods, the motion-compensated IWEs do not include the event cluster associations $P$, which means that sharper object boundaries appear in some IWEs compared to others. The key difference between the EM model in this paper and traditional mixture distribution models lies in the fact that not all motion parameters are mixed. Instead, a specific one-to-one relationship is established between the event $e_{k}$ and the motion neuron $z_j$, resulting in a more precise correspondence.
> > >
> > > Therefore, we can define the joint probability distribution as:
> > >
> > > $$
> > > P(e_k, z_k \mid \theta) = P(z_k \mid \theta) \cdot P(e_k \mid z_k, \theta)
> > > $$
> > >
> > > where the definition of $P(e_k \mid z_k, \theta)$ relates to the IWE value.
> > >
> > > Specifically, we can define the conditional probability of event $e_k$ belonging to motion pattern $z_j$ as:
> > >
> > > $$
> > > P(e_k \mid z_j, \theta) \propto I_j(x_{k_j}' \mid \theta_j)
> > > $$
> > >
> > > where $I_j(x_{k_j}' \mid \theta_j)$ represents the IWE value calculated based on the position and time of event $e_k$ using the parameters $\theta_j\$ corresponding to motion pattern $z_j$.
> > >
> > > We will include this explanation in the revised version of our paper. Thank you again for your valuable comments!

---

> > > > ### Comment · Reviewer_pCA1 · 2024-08-12
> > > > **Thank you very much!**
> > > >
> > > > Thank you for including this explanation in the revised paper.  With that I think I can revisit my review.

---

> > > > > ### Author Response · Authors · 2024-08-13
> > > > >
> > > > > We're deeply grateful for your understanding and your willingness to revisit your review. Your support and consideration mean a lot to us!

---

### Official Review · Reviewer_WNgv · 2024-07-12

**Soundness:** 2
**Presentation:** 1
**Contribution:** 2
**Rating:** 5
**Confidence:** 3

**Summary:**

This paper demonstrates that WTA circuits along with STDP learning resembles EM algorithm-like Bayesian inference and could be used for motion segmentation from event streams by contrast maximization of warped events.

**Strengths:**

The paper proposes an interesting approach for event motion segmentation based on observations from event-based dynamic vision sensors, utilizing a EM-like framework for identifying various motion models from event streams and clustering them into motion patterns. This is achieved using WTA circuits together with STDP-based learning.

**Weaknesses:**

The main weakness of the paper is that the proposed method lacks proper justification of the presented approach, which seems like a heuristic hard clustering method, together with gradient based learning. The experiments also lack depth and the authors demonstrate the high dependence of the performance of the method on the parameter initialization. A more careful writing of the underlying model, the optimization framework and the proposed methodology would be good (see the questions below). Furthermore, the paper lacks more details regarding the choice of $N_{\ell}$ (number of motion models) and the specific forms of the warping functions $W_j$ used. Several steps in the entire methodology, although intuitive, are presented in a heuristic fashion without detailed description and clarity.

**Questions:**

Here are some general questions/comments about the framework:
1. Full form of the abbreviation STDP missing in abstract/introduction.
2. In Eq. (1), why is $\Delta L(x_k,t_k) = q_k\Theta$, since according to line 107, event $e_k$ corresponds to when the intensity change \textit{exceeds} $\Theta$ (noting that $|q_k|=1$). In line 109, add "where $L(x,t)$ is the ... at pixel $x$ \textit{at time $t$}".
3. Line 118: what integration is used? Eq. (2) only describes $I_j(x)$ as a mixture of Dirac measures. Add the definition of $N_e$ (possibly the number of observed events). In Eq. (2), does $x$ represent a pixel? What does the suffix $j$ capture? Based on the description, it suggests that it represents the different \textit{motion models} --  it would be better to explain both the model and the optimization problem in slightly more detail for clarity.
4. The EM framework: while the updates for the model resemble E and M steps in EM, is it actually related? Can you show that this method indeed improves some form of likelihood of the model (recall that EM is most commonly used for MLE in mixture models or other latent variable models)? Can the authors discuss how their method is related to EM (the E and M steps in the current paper are more closer to the hard clustering type algorithms e.g. K-Means rather than EM, particularly the E-step Eq. (5))
5. In Eq. (6), extra $dx$ at the end, also might be better to keep the two terms in a parenthesis.
6. More on the model Eq (2): according to line 119, $p_{kj}$ represents the probability that event $e_k$ belongs to motion model $z_j$, in that case $\sum_j p_{kj}=1$, is that correct? However, in that case, Eq. (2) does not represent a mixture -- i.e., $\sum_k p_{kj}$ might not be 1, can the authors clarify this? Furthermore, the Dirac function in Eq (2) allows the IWE to only pick up values at pixels $x$, where at least one event $e_k$ has been observed (through the transformed position). This does not allow any spatial relation across the pixels - why can the Dirac function not be replaced by some other smooth kernel (like Gaussian)?
7. More on the optimization problem Eq (4): When writing $\text{Var}(I_j)$, whose randomness are we taking the variance (or other expectation operations) with respect to? It seems like $\theta, P$ are parameters (hence fixed) and $x_{kj}'$ is some deterministic transform of the observed events. Can the authors clarify the underlying probability structure of the model?
8. The authors might provide a brief description of the STDP method (and its connections to spiking neural networks) and WTA circuit, which might clarify some of the paragraphs e.g., lines 170-173. It is also unclear why $u_j (t)$ (defined in Eq (10)) is equivalent to $I_j$ (in Eq (2)) as claimed in line 178 - is the $W_j$ in Eq (10) same as the WTA $W$ in Eq (3) -- if so, why is the second input $t_k$ in the latter while $p_{kj}$ in the former?
9. Can the authors explain lines 205-206. It seems like they argue that gradient update increases the variance -- however, this is only the M-step (i.e., conditional on the current values of $p_{kj}$ I am guessing).
10. Why is $u_j$ expressed as a function of time $t$ in Eq (10)? Can the authors clarify how the temporal dependence is captured in the model?

**Limitations:**

See the questions.

---

> ### Author Rebuttal · Authors · 2024-08-07
>
> Thank you for recognizing our **innovative** approach to event motion segmentation using event-based dynamic vision sensors and an EM-like framework. We appreciate your acknowledgment of our method's use of WTA circuits combined with STDP-based learning.
>
> The following are the main issues addressed:
>
> ### **A. Justification of the Approach**
> **1). Theoretical Background:**
> Our research aims to explore the application of Bayesian Inference for motion segmentation by decoupling events using a Spiking Neural Network (SNN)-based framework. This addresses the question, "**How can an SNN model implementing Bayesian computation decouple and segment event streams generated by different motion patterns?**" Building on previous studies [34] that successfully utilized EM-based methods for similar tasks, our work demonstrates both theoretically and experimentally that this method can be implemented using an SNN.
>
> **2). Clarification:**
> We aim to provide a theoretical foundation for applying Bayesian inference using SNNs for event stream decoupling tasks, leveraging the energy efficiency of SNNs. This is particularly relevant for neuromorphic visual sensors and computing hardware (e.g., `Loihi [s1] and SpiNNaker [s2]`), known for their low latency and power consumption. Neuromorphic computing emulates the brain's low-power yet complex visual task-processing capabilities. Previous research validated the hypothesis that SNNs can implement brain-like Bayesian inference. However, its application to neuromorphic computing hardware remains unverified. Our work aims to validate the capability of a spike-based Bayesian computation framework applied to neuromorphic sensors.
>
> **References:**
>
> [s1] M. Davies et al., “Loihi: A neuromorphic manycore processor with on-chip learning,” IEEE Micro, vol. 38, no. 1, pp. 82–99, Jan. 2018.
>
> [s2] Furber, Steve B., et al. "Overview of the SpiNNaker system architecture." IEEE transactions on computers 62.12 (2012): 2454-2467.
>
> ### **B. Detailed Model and Methodology Comprehensive Writing**
> The underlying model, optimization framework, and proposed methodology are detailed thoroughly. We have included pseudo-algorithm code `(please see in the supplementary PDF)` to provide step-by-step explanations, ensuring clarity. This will be added to the revised version for a more comprehensive description.
>
> **Examples and Case Studies:** We have included examples and case studies to illustrate each step of the methodology (Fig. 4-6 in the main text). These practical illustrations help clarify the application of the method and its effectiveness.
>
> ### **C. Choice of Parameters**
> **1). Rationale for $𝑁_l$**:
> The number of $𝑁_l$ can be set based on the experience with the datasets, making sense when trying to separate motion patterns contained in the events. A more intelligent approach can be used after obtaining different motion patterns of other patches. As shown in Fig. 4, clusters of different motions are identified using a non-parametric clustering approach for motion patterns of various patches.
>
> **2). Forms of $𝑊_𝑗$:**
> The forms of $𝑊_𝑗$ depend on the selected motion patterns. A general method uses an affine transformation matrix, but we mainly consider optical flow changes as motion patterns, using $(vx, vy)$ as the motion parameters. Detailed descriptions and justifications for the specific forms of the warping functions (𝑊𝑗) used are provided, explaining their derivation and role in the overall model.
>
> ### **D: Specific Clarifications:**
> **1). STDP, WTA, and Eq. (6):**
> Thank you for the reminder! We will remove dx from equation (6) and add the full name of STDP in the abstract. Descriptions of the STDP method and WTA circuit will be included to clarify related paragraphs.
>
> **2). Line 109 Addition:**
> The formula Δ𝐿(𝑥𝑘,𝑡𝑘)=𝑞𝑘Θ represents changes between consecutive events. Event streams are generated when a threshold is reached, followed by a reset. Thus, the log intensity change between event streams is 𝑞𝑘Θ.
>
> **3). IWE Clarification:**
> We will clarify that IWE results from warping events to the same image plane, where x represents the pixel location. The term "integration" will be removed to avoid confusion. "j" represents the $j-th$ motion model.
>
> **4). EM Algorithm:**
> The EM algorithm, a framework for finding parameters in probabilistic models with latent variables, is used in our approach to iteratively improve motion parameter estimates for event stream motion segmentation. Unlike the Kalman filter, which estimates the state of dynamic systems through prediction and update steps, our approach alternates between the E-step and M-step within the EM framework to achieve event stream motion segmentation.
>
> **5). Var(𝐼𝑗) Explanation:**
> The variance (Var) of the IWE under different motion models shows `(Fig. S1 in the supplemented PDF)` that correct motion models concentrate event distributions at object edges, resulting in higher variance, while incorrect models disperse events, leading to lower variance. This helps in validating the accuracy of motion pattern detection.
>
> **6). Gradient Update Impact:**
> The designed spiking model, combined with the STDP rule, updates motion neuron weights, effectively maximizing the variance of different IWEs, thereby validating the feasibility of the spike-based Bayesian inference framework for event stream motion segmentation both theoretically and experimentally.
>
> **7). Temporal Dependence:**
> The value $u_j(t)$ represents the value of neuron $u_j$ at time $t$. Thank you for pointing it out. We will change this notation in future versions to avoid ambiguity. Our IF spiking neuron model mainly considers input event streams without simulating other temporal decay characteristics.
>
> Thank you again for your valuable feedback. We hope our clarifications and additional details address the concerns raised and further demonstrate the robustness and potential of our proposed method.

---

> > ### Comment · Reviewer_WNgv · 2024-08-13
> >
> > Thank you for the response. Some of the ambiguity have been dealt with and I increase my score to 5. However, although the overall framework is *similar* to EM, it is not clear why it *actually is*. To be precise, it would be better to explicitly write the conditional likelihood to describe the Expectation step. As I mentioned, it seems that it is closer to a hard-clustering algorithm rather than an EM (they are similar but fundamentally different).

---

> > > ### Author Response · Authors · 2024-08-13
> > >
> > > Thank you very much for your feedback and for increasing the score. We appreciate your observation regarding the similarities and differences between our framework and EM.
> > >
> > > To clarify, our method does not implement a hard assignment of events to specific motion patterns. Instead, the probability of an event belonging to a particular motion pattern is proportional to its IWE value (or encoded as membrane potential $u$ in the proposed model) under different motion patterns, as outlined in Eq. (5) and Eq. (11) of the main text. Specifically, the conditional probability $P(e_k \mid z_j, \theta) \propto I_j(x_{k_j}' \mid \theta_j)$, where $I_j(x_{k_j}' \mid \theta_j)$ represents the IWE value associated with the motion pattern $z_j$. This approach allows for a more nuanced assignment that considers the probabilistic contributions from multiple motion patterns rather than a strict, binary assignment.
> > >
> > > In the revised version of the paper, we will explicitly describe the conditional likelihood during the Expectation step to better articulate the similarities with EM, ensuring that these differences are clear and avoiding any further ambiguity.
> > >
> > > Thank you again for your valuable suggestions, which will help us improve the clarity and accuracy of our work!

---

### Author Rebuttal · Authors · 2024-08-07

We sincerely appreciate the valuable feedback from all reviewers. Thank you for recognizing the strengths of our work. Reviewer **WNgv** praised our innovative approach to event motion segmentation using event-based dynamic vision sensors and an EM-like framework, highlighting the use of Winner-Take-All (WTA) circuits combined with Spike-Timing-Dependent Plasticity (STDP)-based learning. Reviewer **pCA1** appreciated our high-resolution motion segmentation using an event-based implementation of the EM algorithm and noted our engagement with the spiking neural network literature. Reviewer **wrk7** recognized the effectiveness of our network for motion segmentation validated on challenging event datasets, including mixed camera self-motion and high-speed moving objects, and commended our spike Bayesian inference framework's interpretability and applicability to neuromorphic vision chips and computing hardware. Lastly, Reviewer **LG4A** endorsed our SNN framework based on WTA as equivalent to the EM algorithm for motion segmentation and supported our application of the Bayesian brain hypothesis using a physiologically interpretable SNN for Bayesian inference on spatiotemporal data from neuromorphic cameras as a promising research direction.

In response to the questions raised, we provide the following clarifications and updates, and have provided some materials and results figures in the uploaded supplementary PDF.

### **A. Theoretical Foundation:**
Our research explores Bayesian Inference for motion segmentation by decoupling events using an SNN-based framework, addressing how SNNs implementing Bayesian computation can decouple and segment event streams generated by different motion patterns. Inspired by previous EM-based methods, we demonstrate the theoretical and experimental viability of this approach with SNNs. We aim to provide a theoretical foundation for applying Bayesian inference using SNNs for event stream decoupling tasks, leveraging their energy efficiency, which is particularly relevant for neuromorphic visual sensors and computing hardware like Loihi and SpiNNaker.

### **B. Model and Methodology:**
Our model, optimization framework, and methodology are thoroughly detailed. We will include pseudo-algorithm code for step-by-step explanations in the revised version for clarity. Examples and case studies illustrate each step of the methodology. We discussed the rationale for setting the number of motion patterns (𝑁ℓ) and provided justifications for the forms of warping functions (𝑊𝑗), explaining their derivation and role. Additionally, we will add the full name of STDP in the abstract and include descriptions of the STDP method and WTA circuit to clarify related paragraphs.

### **C. EM Algorithm Framework:**
In previous work on event stream motion segmentation [34], the M-step focused on optimizing the contrast of different IWEs, effectively projecting the event distribution variance using motion parameters. Our current approach uses the EM framework to iteratively perform the E-step and the M-step to refine the estimates of motion parameters, achieving event stream motion segmentation. Thus, our method does not involve the common concept of maximizing log densities in mixture distribution models.

### **D. Technical Clarifications:**
We clarified various technical details, such as the formula Δ𝐿(𝑥𝑘,𝑡𝑘)=𝑞𝑘Θ, the IWE concept, the use of the EM algorithm, variance explanations for motion models, and gradient update impacts. Our proposed model is not a generative model but is designed to learn motion parameters embedded in event streams generated by different motions, helping segment the event streams according to their respective motion distributions. Implementing this EM-based motion segmentation using SNNs leverages the low-latency and low-power characteristics of neuromorphic hardware, enhancing the performance of event-based cameras and neuromorphic computing systems.

### **E. Supplementary Results:**
We have supplemented our work with further results on object detection based on motion segmentation. Specifically, we calculated the detection success rate on the EED dataset, corresponding to Fig. 6 in the main text. Our detection success rates across three test scenarios are `100%`, comparable to many current state-of-the-art algorithms.

### **F. Parameter Efficiency and Resource Requirements:**
Our method is highly efficient, requiring minimal parameters and computational resources. The model uses neurons corresponding to different motion models and a single global inhibitory neuron to perform WTA. This parameter efficiency and low computational requirement make our approach particularly advantageous for deployment on hardware or neuromorphic hardware, leveraging the low latency and low power consumption characteristics of neuromorphic cameras and computing platforms.

We hope these additions and clarifications address your concerns and further demonstrate the robustness and potential of our proposed method for event-based motion segmentation and object detection using spiking neural networks.

---

### Decision · Program_Chairs · 2024-09-25

**Decision:**

Accept (poster)

**Comment:**

This paper describes a variety of novel applications of Bayesian computation in spiking neural neural computations.  All four reviewers felt that it was above the threshold for acceptance, and I'm pleased to report that it has been accepted to NeurIPS.  Congratulations!  Please revise the manuscript according to the reviewer comments and discussion points, particularly the critical comments of reviewers WNgv and pCA1.